# Improving Rice Quality by Regulating the Heading Dates of Rice Varieties without Yield Penalties

**DOI:** 10.3390/plants13162221

**Published:** 2024-08-10

**Authors:** Jianguo Liu, Qinqin Yi, Guojun Dong, Yuyu Chen, Longbiao Guo, Zhenyu Gao, Li Zhu, Deyong Ren, Qiang Zhang, Qing Li, Jingyong Li, Qiangming Liu, Guangheng Zhang, Qian Qian, Lan Shen

**Affiliations:** 1State Key Laboratory of Rice Biology and Breeding, China National Rice Research Institute, Chinese Academy of Agricultural Sciences, Hangzhou 311401, China; liujianguo316@163.com (J.L.);; 2College of Life Sciences, Zhejiang Normal University, Jinhua 321004, China; 3Chongqing Academy of Agricultural Sciences, Chongqing 401329, China

**Keywords:** Rice (*Oryza sativa* L), flowering time, *OsMADS50*, rice quality, CRISPR/Cas9

## Abstract

The heading date, a critical trait influencing the rice yield and quality, has always been a hot topic in breeding research. Appropriately delaying the flowering time of excellent northern rice varieties is of great significance for improving yields and enhancing regional adaptability during the process for introducing varieties from north to south. In this study, genes influencing the heading date were identified through genome-wide association studies (GWAS). Using KenDao 12 (K12), an excellent cultivar from northern China, as the material, the specific flowering activator, *OsMADS50*, was edited using the genome-editing method to regulate the heading date to adapt to the southern planting environment. The results indicated that the *osmads50* mutant line of K12 flowered about a week later, with a slight increase in the yield and good adaptability in the southern region in China. Additionally, the expressions of key flowering regulatory genes, such as *Hd1*, *Ghd7*, *Ehd1*, *Hd3a*, and *RFT1*, were reduced in the mutant plants, corroborating the delayed flowering phenotype. Yield trait analysis revealed that the primary factor for improved yield was an increase in the number of effective tillers, although there is potential for further enhancements in the seed-setting rate and grain plumpness. Furthermore, there were significant increases in the length-to-width ratio of the rice grains, fat content, and seed transparency, all contributing to an overall improvement in the rice quality. In summary, this study successfully obtained a rice variety with a delayed growth period through *OsMADS50* gene editing, effectively implementing the strategy for adapting northern rice varieties to southern climates. This achievement significantly supports efforts to enhance the rice yield and quality as well as to optimize production management practices.

## 1. Introduction

Rice (*Oryza sativa* L.) is a crucial staple in our nation [1]. The heading time is an important agronomic trait affecting rice’s growth, development, yield, and quality, thereby determining the adaptability and planting regions of rice varieties [2]. Deliberately manipulating the heading times of rice varieties can fully utilize the environmental and climatic resources of the growing region, thereby optimizing the yield potential [3]. Thus, advancing or delaying the heading times of varieties is particularly beneficial for the introduction and popularization of different rice varieties across different regions, especially in the northern and southern areas.

With advancements in biological techniques and in-depth research, an increasing number of genes related to the heading stage have been excavated in rice and *Arabidopsis thaliana*, and these genes are gradually being integrated into a comprehensive flowering signaling pathway [4]. A pair of homologous florigen genes, *HEADING DATE 3a* (*Hd3a*) and *RICE FLOWERING LOCUS T 1* (*RFT1*), is the core of the flowering regulatory network and is regulated primarily by two pathways, *heading date 1* (*Hd1*) and *early heading-date 1* (*Ehd1*), in rice [5,6]. *Hd1* promotes the expressions of *Hd3a/RFT1* under both short-day (SD) and long-day (LD) conditions and regulates the expressions of *Ehd1* and *Hd3a/RFT1* in interaction with *Ghd7* and *DTH8* [7,8]. *Ehd1* regulates the heading stage of the rice by affecting the expression levels of downstream core florigen genes *Hd3a* and *RFT1* under SD and LD conditions [9]. Heading-time genes have pleiotropic effects on yield components [10]. *GL10*, also known as *MADS56*, is a positive regulator of the grain length, grain weight, and heading date, which results in a shorter grain length, a lower grain weight, and delayed flowering in *gl10* plants [11]. *Hd1* and *Ehd1*, two key flowering genes in rice, work together to control the panicle development and, thus, affect the plant yield [12]. *Early flowering-completely dominant* (*Ef-cd*), a long noncoding RNA, positively regulates the expression of *OsMADS50* by mediating the level of histone methylation, thereby promoting the early maturation of the rice without a yield penalty [13]. *RFL*, the homologous gene of *LFY* in Arabidopsis flowering, is the regulator of *OsMADS50*; its expression influences the development of tillers and panicle branches [14]. *DHT3*, also known as *OsMADS50*, causes the heading date of Dianjingyou 1 (DJY1) to be 7–10 days earlier than its near-isogenic line (*dth3*), without significantly affecting the main yield traits, whether grown under short-day (SD) or long-day (LD) conditions [15]. Furthermore, global climate change has exacerbated the threats posed by high temperatures and other extreme weather events to rice production [16,17]. Varieties with appropriate heading stages can maximize their resilience to these stresses, achieving the optimal yield potential. 

Natural variation, as the foundation and driving force for biological breeding, provides new genetic resources for breeding efforts [18]. Genome-wide association studies (GWAS) have been proven to be an effective new strategy for explaining the genetic basis of complex traits, with the advantage for improving the efficiency for detecting natural variations [19,20]. In fact, most GWAS studies focus on dissecting the genetic basis of individual yield traits by analyzing the relationship between nucleotide polymorphisms and phenotypic variations using different population sets [21,22]. So far, GWAS has been successfully applied to dissect complex traits in multiple crop species, particularly in the study of *AP2/ERF* and *HST1*-*like* genes, demonstrating its importance in natural variation research [23,24,25,26].

Gene-editing technology is a powerful technology to create new varieties and has great potential in breeding species [27]. Thanks to the development of gene-editing technology, a large number of studies on heading-date gene editing have shown the close connections between heading dates and yield-related traits. The optimal balance between the growth period and yield can be achieved to obtain new varieties with better adaptabilities and higher yields [17,28]. By down-regulating the expression of *Ehd1*, a key nodal gene in the heading date network, using gene-editing technology, rice varieties with slightly delayed heading dates and higher yields were created [29]. Li et al. generated different weak Ehd1 alleles in japonica rice by editing the *Ehd1* promoter region and identified a rice variety, which exhibited delayed flowering and an increased yield potential, that be suitable for breeding purposes [29]. The open reading frame was edited to fine-tune the expression of *Hd2* and adjust the heading date of the variety by the CRISPR/Cas9 genome-editing system [30]. Knocking out the late-flowering gene *OsGhd7* by CRISPR/Cas9 can make rice mature early in multiple geographical locations in China [8]. Using gene editing, the researchers analyzed the regulatory modules of the core flowering genes *Hd1*, *Ghd7*, and *DTH8* under the LD and SD modules, as well as the adaptive distribution of the rice [7].

In this study, we identified a major locus that regulates the heading date in rice, through genome-wide association studies. The northern superior variety, K12, characterized by its high yield, yield stability, high quality, strong resistance to diseases, and tolerance to flooding, was selected for introduction. To evaluate the yield potential and enhance the regional adaptability of K12 in southern regions, including Hangzhou, we utilized CRISPR/Cas9 genome-editing technology to knock out the flowering regulation gene *OsMADS50* in the K12 background. The heading times of the new *osmads50* lines were delayed by about one week, with modest increases in the yields. This study explored the practical application of *OsMADS50* in regulating the heading stage of the rice variety K12, which provided a basis for the broader adaptation of the rice.

## 2. Results

### 2.1. Identification of Significant Loci for Heading Dates Using GWAS

We have undertaken a study on the heading date trait of rice based on the data results of 3021 rice varieties obtained through the investigation and analysis of data on the RiceSuperPIRdb-3K rice database website (http://www.ricesuperpir.com/web/tools (accessed on 1 March 2023) [31,32]. The days to heading ranged from 64 to 148, with an average of 95 in all the accessions, the GJ group, and the XI group. The heading date traits displayed approximately normal distributions in all three groups (Figure 1A–C). The GWAS analysis results of 3k rice materials under all the accessions and the GJ-group and XI-group conditions indicate that significant SNPs, with a threshold (−log10 (P)) of higher than 6, are predominantly concentrated on chromosomes 3 and 10 (Figure 1D–F and Appendix A). We confined all these significant SNPs to five QTL genomic regions, with about 400 kb, named q1-q5. The genes in the five QTL regions were scanned based on the Rice Genome Annotation Project (http://rice.uga.edu/cgi-bin/gbrowse/rice/ (accessed on 3 March 2023) to search for genes involved in heading dates. A total of 59 annotated genes were identified, including 1, 1, 6, 26, and 25, in q1-q5, respectively (Table 1). Among them, there are five known functional genes; *Hd1* and *OsMADS50* are star genes regulating the heading stage; *RSR1* and *FLO7* are related to endosperms, and *FLO7* is related to chloroplast development; and only *OsMADS50* shows significantly associated SNPs in all three groups (Table 1). *OsMADS50* is an MADS-box transcription factor gene that can promote flowering in rice through overexpression and delay flowering by inhibiting its expression [33]. Previous studies have shown that the expression of *OsMADS50* affects the flowering regulatory network, the development of tillers and panicle branches, and the development of rice crown roots [13,14,15,33,34,35], suggesting that *OsMADS50* plays important roles in rice growth and development and adaptation to environmental changes and has potential application values in rice molecular breeding and crop production.

### 2.2. Phylogeny Analysis of OsMADS50

To study the conservation of the protein OsMADS50 across different species and its evolution in the process of the species evolution, we utilized the structural domain prediction from the SMART website (http://smart.embl-heidelberg.de/ (accessed on 20 March 2023) and found that OsMADS50 contains a keratin-like (K-box) domain (Appendix A). Additionally, the OsMADS50 protein belongs to the MADS50 family of proteins. Subsequently, we used the HMMER sequence identifier (PF01486) to search for, remove redundant and incomplete sequences from, validate the domain for, and obtain members of the rice’s MADS protein family.

The evolutionary tree constructed with 69 rice, seven wheat, and three Arabidopsis thaliana MADS protein sequences shows that the MADS genes in rice and other species can be divided into six subfamilies based on sequence similarity, named subfamilies A-F. The branching between the subfamilies is clear, indicating an early differentiation of rice MADS genes. Subfamilies D and F contain the most rice MADS family members, accounting for 20.28% and 21.74%, respectively; subfamily C has the fewest rice MADS family members, only accounting for 13.04% (Figure 2). In subfamily F, the flowering-related proteins from Arabidopsis, AT5G51860 [36], AT2G45660 [37], and AT5G62165 [38], are direct orthologs of rice MADS proteins related to the heading stage regulation, further validating the accuracy of the evolutionary tree results. Notably, the wheat proteins, Traes 6BS 43B59D772 and Traes 6AS D6ABA1D79, highly homologous to OsMADS50, respond to abiotic stress during growth and development, suggesting a potential stress-resistance function for OsMADS50 [39].

### 2.3. Protein Structural Analysis of OsMADS50

Protein domains can provide crucial information for studying the biological functions they perform. We conducted a conservation motif analysis on 12 rice MADS protein members of the F subfamily highly homologous to OsMADS50 using the MEME website (http://meme-suite.org/ (accessed on 20 March 2023) and TBtools. The results showed consistency between the evolutionary tree conservation analysis and the distribution of the conserved motifs and revealed a total of eight conserved motifs (Figure 3). We also found that all the members of the F subfamily contain Motif 3, leading us to speculate that Motif 3 is closely related to the K-box domain. In comparison to other proteins, the target protein OsMADS50 uniquely exhibits strong specificities for Motif 3, Motif 4, and Motif 7, while lacking the Motif 1 and Motif 8 structures found in many other members (Figure 3). This indicates significant differences in the conserved motifs between protein OsMADS50 and other MADS proteins, foreshadowing the functional diversity of OsMADS50 and suggesting further exploration potential.

### 2.4. osmads50 Mutants’ Delayed Growth Period

*OsMADS50* functions as an LD-specific flowering activator, and in the *OsMADS50* pathway, the *osmads50* mutants had delayed heading dates without yield penalties [35,40]. We selected variety K12, a high-quality japonica rice variety that matures early in the second temperate region of Heilongjiang Province and has a large area of promotion, which also belongs to a long-day region, for further experiments. We designed the target site and constructed a CRISPR/Cas9 vector that targets *OsMADS50* (Figure 4A) and transformed the K12 background to obtain transgenic plants. In the T_0_ generation, we sequenced and genotyped each positive plant and selected homozygous mutants for the next experiment (Figure 4A). The homozygous mutant lines of the T_1_ generation were chosen for growth period statistics. The statistical results show that the flowering times of the mutants were significantly later than the K12 variety during the growing period in Hangzhou, a region known for its naturally long daylight hours (Figure 4B,C). To further analyze the network changes that regulate flowering, we simply analyzed the expressions of related genes by RT-qPCR. RT-qPCR assay results indicated that in *osmads50* mutants, the transcript levels of *OsMADS50*, *Ef-cd*, *Ehd1*, *Hd3a*, and *RFT1* were lower than those of the wild type under long daylight conditions (Figure 4D). Studies have shown that the expression of *Ef-cd* was positively correlated with that of *OsMADS50* [13]. The expression levels of *OsMADS50* and *Ef-cd* showed the same trend, which was consistent with the expectation and verified the reliability of the quantitative results. *OsMADS50* acts upstream of *Ehd1*, *Hd3a*, and *RFT1*, and it regulates the flowering process in response to long daylight conditions [41]. The down-regulation of the expression of *OsMADS50* led to the down-regulation of the expressions of the three corresponding downstream genes and finally led to the delay in flowering.

### 2.5. osmads50 Mutants’ Significantly Increased Tiller Numbers

As is known, a longer growth period is associated with increased yields and adaptabilities; also, studies have shown that a delayed growth period can be weakly correlated with the rice yield, plant height, grain weight, spikelet number per panicle, panicle length, and seed-setting rate [29]. We quantified these effects among K12 and *OsMADS50* function-loss mutant lines of K12; the statistical results show that the mutants of K12 had significantly increased effective tillers and grain lengths than the K12 plants, and there were no significant changes in the plant height, grain number per five main panicles per plant, and yield per plant between the *osmads50* mutants and wild type (Figure 5A,D–H,K). It is worth mentioning that the yields per plant of *osmads50* (*m1* and *m2)* were slightly increased compared with that of the wild type (Figure 5G). The statistical results of the other key agronomic traits of the yield showed that the 100-grain volumes, the 1000-grain weights, and the grain widths of *m1* and *m2* were significantly lower than those of K12 (Figure 5I,J,L), while the secondary branch number per main panicle and grain number per plant of *m1* and *m2* were higher than those of K12 (Figure 5B,C and Appendix A). Considering the three key factors of the rice grain yield (the number of effective tillers per plant, the number of filled spikelets per panicle, and the grain weight), we speculated that the limiting factors of the yields of *osmads50* mutants might be the seed-setting rate and grain fullness.

### 2.6. Improved Appearance Quality of osmads50 Seeds

The length-to-width ratio of rice grains is one of the important agronomic traits that affect the quality and yield of the rice [42]. Generally speaking, an increase in the length-to-width ratio improves the appearance quality, although in some special cases, a decrease in the length-to-width ratio can also improve the appearance quality [43]. We found that compared to the K12 wild type, the *m1* and *m2* mutant plants showed significant increases in the length-to-width ratio; the transparency of the mutants was enhanced (Figure 6A,B), and both the percentage of chalky grains and chalkiness degree of the *osmads50* mutant were significantly reduced (Figure 6C,D). These results indicated an improvement in the appearance quality of the *osmads50* mutant, consistent with the general understanding that finer grains are associated with a lower chalkiness degree.

### 2.7. Slightly Modified Physicochemical Properties of the osmads50 Seeds

Starches (from 85% to 90%), proteins (from 7% to 12%), and lipids (from 0.3% to 3%) are the primary components affecting the cooking and eating qualities of rice [44,45]. To further evaluate the eating and cooking qualities of the *osmads50* mutant, the main physicochemical properties of the rice seeds were investigated. In general, the mutant’s physicochemical properties were either unchanged or exhibited only minor variations, and the contents of amylose and protein in the *osmads50* mutant were comparable to those found in the K12 control (Figure 7D,E). Particularly, the fat content in the mutant significantly increased (Figure 7F). The alkali spreading value results show that the alkali spreading value of K12 is around 6.2, while those of *m1* and *m2* are both around 6.7 (Figure 7A–C). The slightly higher alkali spreading value of the *osmads50* mutant indicates a slight decrease in the gelatinization temperature of the *osmads50* mutant. These data suggest that the *OsMADS50* mutation has little effect on the contents of amylose and protein in rice while significantly increasing the fat content.

### 2.8. Haplotype Analysis of OsMADS50

The *OsMADS50* gene comprises six exons and five introns. Utilizing the Rice Functional Genomics and Breeding website (https://www.rmbreeding.cn/Index/ (accessed on 2 February 2024), an analysis of this gene and its upstream 2 kb promoter region was conducted [46]. The analysis indicated that there are 24 non-synonymous SNP locus mutations, with 10 located in the 3’UTR region and 14 in the promoter region (Figure 8A). Utilizing the SNP sites present in the 3’UTR region of the *OsMADS50* gene, the 3000 rice varieties stored in the database were mainly divided into three haplotypes: Hap1 (GTTGCCAACC), Hap2 (ATATTCTCTC), and Hap3 (AATGTGACTG). Hap1 predominantly comprises about 91% of the japonica rice varieties, while Aus and Bas rice varieties constitute 7% and 2%, respectively (Figure 8B). Within Hap2, 87% of the varieties are indica rice, whereas japonica rice and basmati rice make up 6% and 7%, respectively (Figure 8B). Hap3 predominantly includes 93% of the indica rice varieties (Figure 8B). These results indicated a strong correlation between the differential haplotypes in the 3’UTR region of this gene and the differentiation between japonica and indica rice varieties. Similarly, the 14 SNP sites located in the promoter region of this gene served to categorize the 3000 rice varieties in the database into three haplotypes: Hap1 (AGGATTCGTGAATG), Hap2 (GTGAACTACGAGAT), and Hap3 (GGTTACCGCAGGAG). Within Hap1, approximately 88% of the varieties are japonica rice, followed by Aus and Bas rice varieties (Figure 8C). In Hap2, more than 90% of the varieties are indica rice. As for Hap3, 87% of the varieties are indica rice, succeeded by japonica rice and Bas rice varieties (Figure 8C). In conclusion, the haplotype differences of this gene mainly correlate with the distinctions between japonica and indica rice varieties.

## 3. Discussion

Although previous studies have shown many cloned heading date genes, most of them were major genes that are challenging to apply in rice breeding [1,35,47,48]. In our study, we identified the *OsMADS50* locus by GWAS using rice populations collected from 3021 rice varieties (Figure 1A). Meanwhile, the analysis identified that SNPs located in the *OsMADS50* gene region were significantly associated with the heading dates in rice across different varieties (Figure 1D–F and Figure 8A,B). In the introduction process, some varieties are not adapted to the differences in the regional temperature and photoperiod, resulting in certain less favorable agronomic traits [4,49]. In our study, although the yield per plant increased, the 1000-grain weight and grain filling decreased significantly (Figure 5G–I). It is worth mentioning that the number of tillers of *m1* were significantly higher than that of K12 (Figure 5F); the increase in the tiller number is the key factor of the high and stable yields of the *osmads50* mutants. Therefore, without reducing the rice yield, the optimization and adjustment of the growth period are crucial to enhance the adaptability of rice varieties and expand their planting areas.

Many heading-date- and yield-related studies have been reported, and some have shown that the heading date was negatively correlated with the yield [16,40,50,51]. However, within a certain growth cycle, excessive delay in the flowering time would lead to yield reduction [52,53]. Previous studies have shown that the expression levels of florigen genes *Hd3a* and *RFT1* were highly correlated with the heading stages of cultivated rice [6,54,55,56]. In this study, the expression of the flowering-determining gene *Hd3a/RFT1* was reduced in the *osmads50* mutants, resulting in about a week of delayed flowering and increased yield production (Figure 4C,D). In the late-flowering mutant *lvp1*, the expression of *OsMADS50* was down-regulated [57]. These findings were consistent with our expectations. In addition, our expression results also showed that the expression levels of four flowering time genes *OsGI*, *Hd1*, *Ghd7*, and *DTH8* were decreased (Appendix A). The network that regulates flowering is complex, and although not in the same pathway, or parallel pathways, it can also lead to changes in the expressions of other related genes in heading date pathways [7,58]. 

*RFL* is a regulator factor of the flowering activator *OsMADS50*; the knockout of the *RFL* resulted in a severe delay in flowering; a severe phenotype, even without any heading; and reduced tillering and branching [14]. We found that *RFL* gene expression was not significantly different among wild type and *osmads50* plants (Appendix A). *Hd3a* also acts as a mobile signal promoting rice branching: Rice *Hd3a* RNAi plants flowered about 20 days later than the wild type and were accompanied by reduced numbers of branches [59]. However, the number of tillers in the *osmads50* plants increased significantly in our results (Figure 5F), suggesting that the *OsMADS50-Ehd1-Hd3a/RFT1* pathway under LD conditions regulating plant branching is complex and involves many other transcription factors or proteins. *OsMADS34* is a positive regulator of the formation of the spikelet meristem, which affects the development of rice panicles [60], and its mutant increased the number of primary and secondary branches and reduced the setting rate [61]. Our results indicate decreased expression of *OsMADS34* in *osmads50* plants, exhibiting a phenotype similar to *osmads34*, which showed an increase in the number of primary and secondary branches and a decrease in the setting rate (Figure 5C and Appendix A and Table 1). In conclusion, increases in the numbers of branches and tillers are the keys to the slight increase in the rice yield of *osmads50* plants in K12, and whether this trend is applicable to other rice varieties remains to be verified.

Although a number of studies have revealed the functions of OsMADS50 in various stress responses and plant growth, its roles in regulating the rice quality and germination traits have not been reported so far [34,35,41,55]. In this study, we found that the conservation of the protein OsMADS50 across the different species and its evolution during species evolution respond to biological functions during growth and development, suggesting potential roles in regulating the rice quality and germination traits (Figure 3 and Figure 5). The rice grain quality typically encompasses cooking, eating, appearance, and nutrition [42]. Starch, seed storage proteins, and lipids determine the rice grain functional properties, including the gelatinization temperature, gelatinization index, flour viscosity, and rapid viscos analyzer (RVA) pasting properties [44]. We observed that compared to the wild type, the *OsMADS50* mutant showed a significant increase in the length-to-width ratio, a higher fat content, and enhanced seed transparency (Figure 6A,B and Figure 7F). Our study helps to elucidate the roles of *OsMADS50* in controlling various aspects of rice yield and quality traits.

In this study, we identified *OsMADS50* as a major locus regulating the heading stage of the rice. We created new *osmads50* mutant lines of K12 with a delayed heading date of about one week and increased the yield per plant with an improved rice appearance quality. We also found that there is still potential for enhancing the rice yields of the new lines as the grain filling remains incomplete. Also, we can use gene-editing technology to edit the promoter region of *OsMADS50* to create more lines with fine-tuned gene expressions and obtain varieties with more heading phenotypes.

## 4. Materials and Methods

### 4.1. Genome-Wide Association Analysis (GWAS)

Using a mixed linear model (MLM), a population structure, and a phylogenetic matrix (Q + K) as covariables, we conducted a genome-wide association analysis on the heading date phenotypic traits of 3021 rice varieties on the RiceSuperPIRdb-3K rice database website (http://www.ricesuperpir.com/web/tools (accessed on 1 March 2023).

### 4.2. Plant Materials and Growth Conditions

The japonica variety Kendao12 (K12) is from Heilongjiang Province, China. All the plants and their offspring were grown in a greenhouse in winter (at 30 °C in 16 h of light and 28 °C in 8 h of darkness). In summer, all the rice plants were cultivated in the paddy fields at the China National Rice Research Institute in Hangzhou. 

### 4.3. Phylogenetic Analysis and Protein Structural Analysis

Using the ClustalW program in MEGA7 software (7.0.26), the K-box domain was subjected to multiple-sequence alignment [62]. A phylogenetic tree was then constructed using the neighbor-joining method, with a bootstrap value of 1000 and the Poisson model selected as the model. All the other parameters were set at their default values.

The conserved motifs of the K-box domain of the rice MADS50 and other species were identified using TBtools software (v2.056). The motif number was set at 12, and all the other parameters were set at their default values. The annotations of the identified motifs were obtained from the InterPro database (https://www.ebi.ac.uk/interpro/ (accessed on 20 March 2023). The protein sequences were submitted to the HMMER database (https://www.ebi.ac.uk/Tools/hmmer/search/hmmscan (accessed on 3 March 2023)) to analyze their conserved domains. TBtools software (v2.056) was used to generate a schematic diagram of the conserved motifs and domains of the protein.

### 4.4. Plasmid Construction and Agrobacterium-Mediated Rice Transformation

The genomic DNA sequences of *OsMADS50* were acquired from the Rice Genome Annotation Project (RGAP; http://rice.uga.edu/cgi-bin/gbrowse/rice/#search (accessed on 20 March 2023). The sequence of TATTGAAGAACTGCATAGCCTGG was selected as the target site for the knockout of *OsMADS50* using the CRISPR/Cas9 system. The construction method of the expression plasmid was as previously described [63]. The constructed plasmid was transferred into K12 using the *Agrobacterium*-mediated transformation (strain *EHA105*) method. The primers are listed in Appendix A.

### 4.5. Detection of Mutations and Transgenic-Free Plants

Genomic DNA was extracted from rice leaf tissues of mutant plants via the CTAB method. PCR was conducted with 2 × Phanta Flash Master Mix (Vazyme, P510-02, Nanjing, China) to amplify the genomic regions containing the target site. The amplified fragments were sequenced using the Sanger method and decoded using the degenerate sequence-decoding method [64]. PCR amplification was performed using primers Hyg-F1/Hyg-R1 and Cas9-F/pC1300-R, which target the Hygromycin phosphotransferase (HPT) and Cas9 cassettes, respectively. The PCR primers used in this study are listed in Appendix A.

### 4.6. RNA Extraction and Quantitative Real-Time PCR (qRT-PCR)

Fresh leaves of the K12 and *osmads50* mutants, at heading stage, were stored with liquid nitrogen to extract the total RNA. The total RNA of the samples was extracted with RNA extraction kits (Axygen, AP-MN-MS-RNA-250, Union City, CA, USA), and the first-strand cDNA was reverse transcribed using MonScript™ RTIII Super Mix with a dsDNase (two-step) kit (Monad, MR05201M, Wuhan, China). The RT-PCR experiment was performed using the Taq Pro Universal SYBR qPCR Master Mix (Vazyme, Q712-02, Nanjing, China) with three biological replicates. Gene expression was measured using qRT-PCR and the actin gene as an internal control. The qRT-PCR primers for actin, *Hd1*, *Ghd7*, *DTH8*, *Ehd1*, *Hd3a*, *RFT1*, *OsGI*, and *OsMADS50* are listed in Appendix A. 

### 4.7. Measurement of Rice-Yield-Related Traits

The plant heights and tillers of K12 and *osmads50* plants were measured during the maturing stage of plants grown in the transgenic experimental field. The grain and panicle traits of the rice were measured in the laboratory. Mature rice seeds from superior rice panicles were collected and air-dried, and the rice grain length, grain width, grain weight and yield per plant were measured using a rice appearance quality detector (SC-E, Wanshen, Hangzhou, China). Data are shown as means ± s.e.m. (n ≥ 5). Student’s *t*-test was used to assess the significance of the differences between the means.

### 4.8. Evaluation of Rice Appearance Quality

First, naturally dried mature rice grains were dehusked using a huller (OTAKE, Dazhu, Japan). Then, the regular brown rice was hand-selected and polished with a Kett grain polisher (Tokyo, Dazhu, Japan). Finally, the chalkiness trait of the milled rice seeds was assessed utilizing a Wanshen SC-E rice seed detector.

### 4.9. Determination of Alkali Spreading Value in Rice

A 1.7% KOH solution was prepared. Twelve whole polished rice grains that were unbroken, uncracked, uniformly sized, and mature were selected and placed in a clean Petri dish (or a square box), ensuring replication. Then, 2 mL of 1.7% KOH was added to each dish, and the rice grains were evenly spread, leaving sufficient gaps between each grain to facilitate decomposition and diffusion. The dishes were securely covered. The Petri dishes were carefully placed in a (30 ± 0.5) °C incubator and allowed to stand for 23 hours until the rice grains fully disintegrated. The degrees of digestion and diffusion of the rice grains’ endosperms were visually observed grain by grain [65].

### 4.10. Physicochemical Analyses of Rice

Approximately 10 g of mature and plump whole rice grains was randomly selected for the detection of the protein content, fat content, and amylose content by combining near-infrared spectroscopy and chemometrics using a near-infrared analyzer (Foss DS2500F, Denmark, Nordborg) [66].

## 5. Conclusions

In this study, our results demonstrate that identified single nucleotide polymorphisms (SNPs) located in the *OsMADS50* gene region are significantly associated with the heading dates of different rice varieties, through genome-wide association analysis. Further, we utilized CRISPR/Cas9 gene-editing technology to genetically modify an excellent cultivar from northern China, KenDao 12 (K12), successfully cultivating a new rice line based on K12. These new lines significantly delay the heading date by approximately one week and effectively enhance the yield through an increase in the number of effective tillers. Interestingly, these lines also significantly improve the length-to-width ratio of the rice grains, increase the fat content, and significantly enhance the seed transparency, all of which contribute to the overall improvement in the rice quality. This research not only reveals the important role of the *OsMADS50* gene in regulating the heading date of the rice but also provides new insights and methods for rice variety improvement. 

## Figures and Tables

**Figure 1 plants-13-02221-f001:**
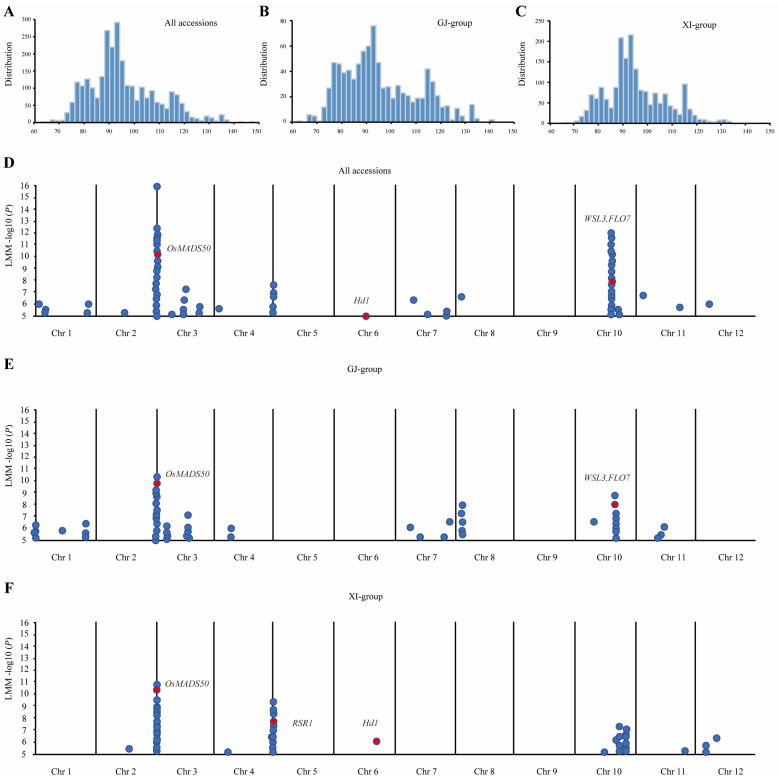
GWAS for heading dates in 3k rice accessions. (**A**) Distribution of heading dates in all accessions. (**B**) Distribution of heading dates in GJ group. (**C**) Distribution of heading dates in XI group. (**D**) Manhattan plot schematic diagram of GWAS for heading dates in all accessions. (**E**) Manhattan plot schematic diagram of GWAS for heading dates in GJ group. (**F**) Manhattan plot schematic diagram of GWAS for heading dates in XI group. The dots represent the strength of the association between SNPs and the heading date phenotype.

**Figure 2 plants-13-02221-f002:**
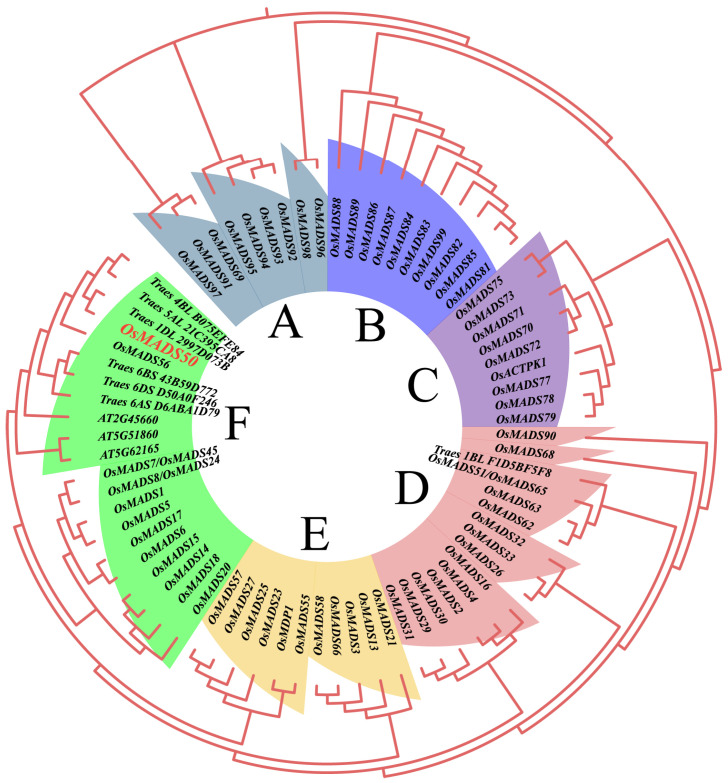
Phylogenetic analysis of OsMADS50 proteins. The six colors represent six different subfamilies.

**Figure 3 plants-13-02221-f003:**
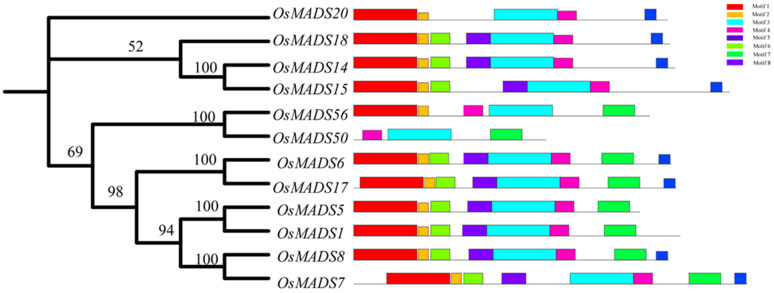
Motif patterns of OsMADS50 proteins. The numbers refer to the node support, and the higher the node support, the more reliable the evolutionary relationship of the biological taxon represented by that node.

**Figure 4 plants-13-02221-f004:**
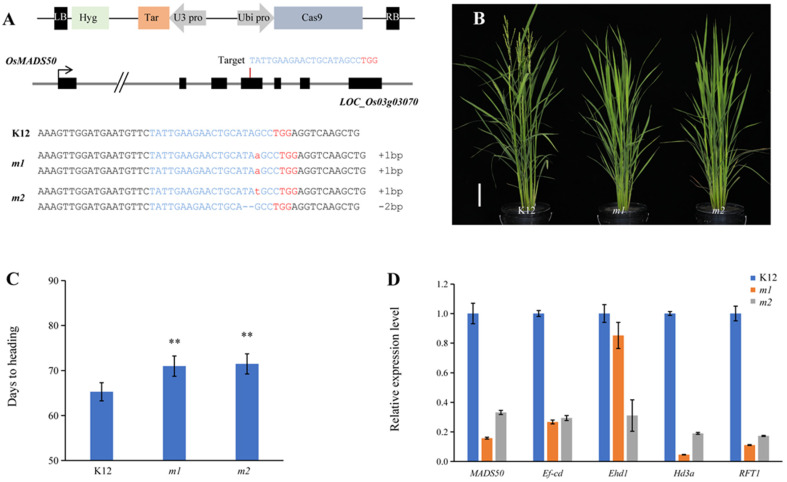
*osmads50* mutants have a delayed growth period. (**A**) Schematic diagram of the targeted sites in *OsMADS50.* Black boxes represent exons of *OsMADS50,* and the arrow indicates the direction of translation. The targeted sequence is highlighted in blue, and the protospacer adjacent motif (PAM) sequences are marked in red. (**B**) The plant morphologies of K12 and the *osmads50* mutants of K12. Bars = 10 cm. (**C**) Days to heading of K12 and *osmads50* mutants of K12; n ≥ 10. Error bars represent standard deviation (SD). ** *p* < 0.01 (Student’s *t*-test). (**D**) Relative expression levels of *MADS50*, *Ef-cd*, *Ehd1*, *Hd3a*, and *RFT1* in K12 and *osmads50* mutants of K12. Error bars represent standard deviation (SD).

**Figure 5 plants-13-02221-f005:**
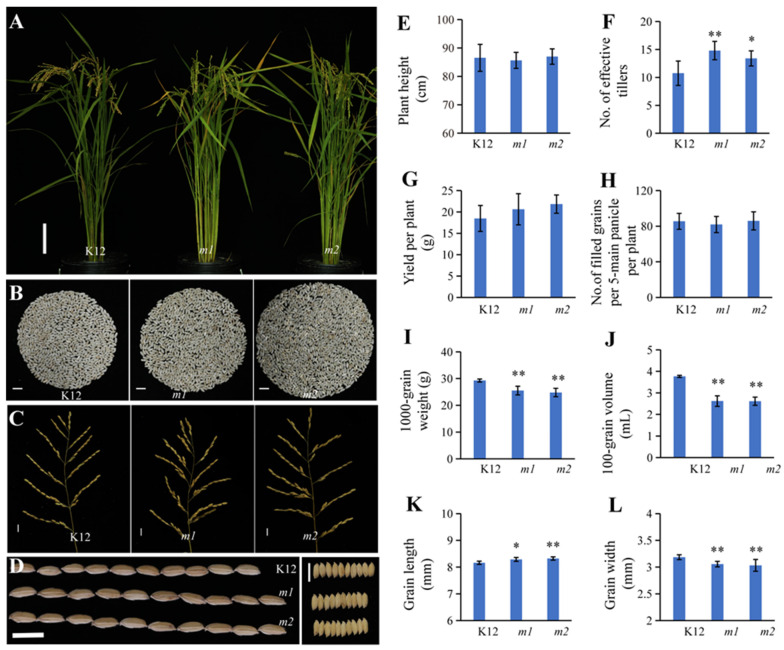
Yield-related trait characterizations of wild type and *osmads50* mutants of K12. (**A**) Plant morphologies of K12 and *osmads50* mutants of K12 at maturity of K12. Bar = 10 cm. (**B**) The morphologies of the brown rice grains per plant among K12, *m1,* and *m2*. Bars = 1 cm. (**C**,**D**) The morphologies of the main panicles and the grain shapes among K12, *m1*, and *m2.* Bars = 1 cm. (**E**) Comparison of plant heights among plants of K12, *m1*, and *m2*. (**F**) Comparison of number of effective tillers among K12, *m1*, and *m2*. (**G**) Comparison of number of filled grains per five main panicles among K12, *m1*, and *m2*. (**H**–**L**) Comparisons of 1000-grain weights, 100-grain volumes, grain lengths, and grain widths of K12, *m1*, and *m2*. Error bars represent standard deviation (SD); * *p* < 0.05; ** *p* < 0.01 (Student’s *t*-test).

**Figure 6 plants-13-02221-f006:**
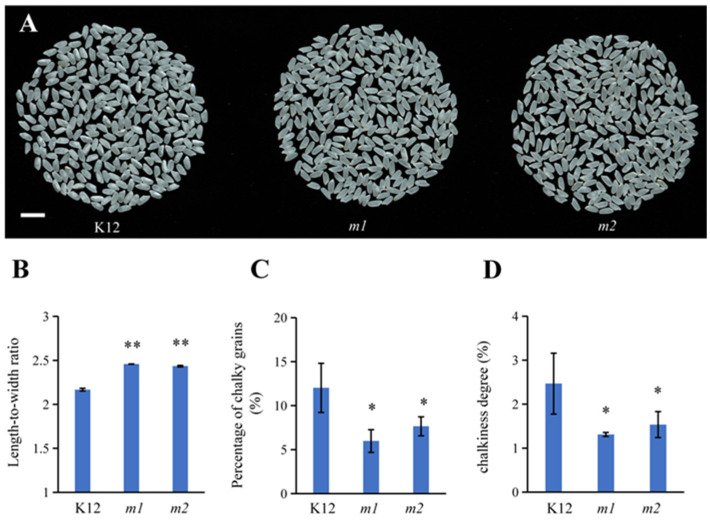
Analysis of the appearance qualities of the *osmads50* mutant and the K12 control. (**A**) Comparison of the appearances of polished rice among K12, *m1*, and *m2*. Bar = 1 cm. (**B**) Length-to-width ratios of K12, *m1*, and *m2*. (**C**) Percentages of chalky grains among K12, *m1*, and *m2*. (**D**) Chalkiness degrees of K12, *m1*, and *m2*. Error bars represent SD; ** *p* < 0.01; * *p* < 0.05 (Student’s *t*-test).

**Figure 7 plants-13-02221-f007:**
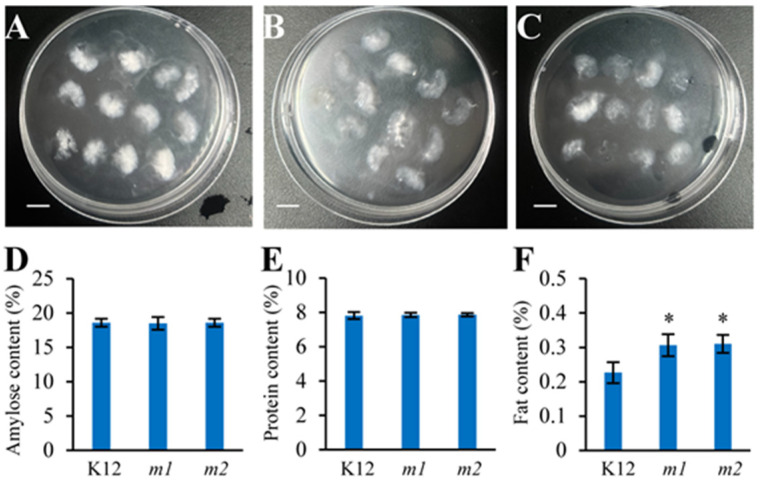
Physicochemical analysis of the *osmads50* mutants and the K12 control. (**A**–**C**) Alkali spreading values of the *osmads50* mutant and the K12 control: A, K12; B, *m1*; C, *m2*; bars = 1 cm. (**D**) Amylose contents of the *osmads50* mutants and the K12 control. (**E**) Protein contents of the *osmads50* mutants and the K12 control. (**F**) Fat contents of the *osmads50* mutants and the K12 control. Error bars represent SDs; * *p* < 0.05 (Student’s *t*-test).

**Figure 8 plants-13-02221-f008:**
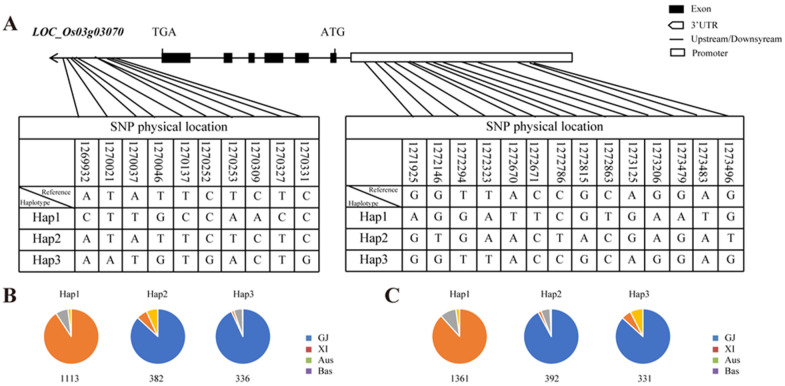
Haplotype analyses of *OsMADS50*. (**A**) Schematic representation of *LOC_Os03g03070* structure and the positions of 10 and 14 SNPs in3’UTR and promoter, respectively, used for haplotype analysis. (**B**,**C**) Distributions of three major haplotypes in 583 rice accessions. Numbers of detected haplotypes are given below each panel.

**Table 1 plants-13-02221-t001:** GWAS regions associated with heading dates.

QTL	Chr.	Physical Region (nt)	Significant	Lead SNP		Group	Co-Location Loci
			SNPs	Position (nt)	LMM-log10(*P*)		
q1	6	9,138,220–9,538,220	1	9,338,220	7.29	All; XI	*Hd1*
q2	5	960,013–1,360,013	1	1,160,013	7.99	XI	*RSR1*
q3	3	1,070,331–1,470,331	6	1,270,331	10.63	All; GJ; XI	*OsMADS50*
q4	10	16,609,046–17,021,303	26	16,809,046	8.82	All	*/*
q5	10	17,029,581–17,429,581	25	17,228,554	7.71	All; GJ	*WSL3; FLO7*

## Data Availability

The data supporting the findings of this study are available within the article or its Appendix A.

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
