# Peer review of "Improving Rice Quality by Regulating the Heading Dates of Rice Varieties without Yield Penalties"

_plants, 2024, doi:10.3390/plants13162221_

Round 1

Reviewer 1 Report

Comments and Suggestions for Authors

The manuscript, “Improving rice quality by regulating the heading date of rice varieties without yield penalties”, reports the identification of a rice flowering/heading time regulator (i.e. OsMADS50) using a genome wide association study. Based on these findings, the authors mutated the OsMADS50 gene in KenDao12 (K12) cultivar using CRISPR/Cas9 strategy and performed molecular and morphological analyses to elucidate the regulatory mechanism. In addition, the authors conducted yield trait and grain quality analyses and demonstrated its roles in rice quality improvement. The OsMADS50 gene has been studied in regulating long day specific flowering/heading time for two decades, and its molecular function has been finely characterized (Lee et al. 2004, Plant Journal). Similar experiments/results, such as examining the expression profile of flowering time related genes in osmads50 background, have also been reported (Ryu et al. 2009, Plant Cell & Environment). Thus, the novelty and scientific sound of current study are limited. Although the design and analysis of this study are correct and the results are detailed, some results are required to be improved. Below, I outlined some of my major concerns that need to be addressed before next submission.

Generally, the language of this manuscript is required to be carefully examined. 

INTRODUCTION:

The authors introduced the importance of heading time in rice growth. I suggest that they should also delineate the connection between heading time and rice yield (and biomass). Also, it is worth to mention that optimizing heading time could assist crops in adapting to adverse environmental stresses.

I suggest the authors citing more up-to-date reference to enhance the significance of the current study.

RESULTS:

The procedure of GWAS should be detailed.

The candidate genes could be labeled in figure 1D-E.

Figure 4A, the m2 does not appear to be a homozygous line. The primers used for detecting the mutation could be labeled in the fig4A.

Section 2.8, is there any SNP found in the coding region of OsMADS50?

I am interested to learn that if the SNPs identified in the promoter region (2k bp upstream of start codon) of OsMADS50 gene lead to the mutation in any important cis-regulatory elements. If so, how does the mutation affect the expression of OsMADS50 in different natural varieties?

DISCUSSION:

Based on previous study, the OsMADS56 physically interacts with OsMADS50, and plays an antagonistic role with OsMADS50 in regulating rice flowering time (Ryu et al. 2009, Plant Cell & Environment). Thus, a detailed investigation of OsMADS56 should be conducted under the osmads50 background, to explore if the yield and grain quality improvement is due to the potential irregular expression of OsMADS56.

MATERIALS AND METHODS:

The procedure of GWAS is missing.

Section 4.6 to 4.9 should be described in details.  

Comments on the Quality of English Language

Moderate editing of English language required

Author Response

Responses to Reviewers

Reviewer #1:

Comments and Suggestions for Authors

The manuscript, “Improving rice quality by regulating the heading date of rice varieties without yield penalties”, reports the identification of a rice flowering/heading time regulator (i.e. OsMADS50) using a genome wide association study. Based on these findings, the authors mutated the OsMADS50 gene in KenDao12 (K12) cultivar using CRISPR/Cas9 strategy and performed molecular and morphological analyses to elucidate the regulatory mechanism. In addition, the authors conducted yield trait and grain quality analyses and demonstrated its roles in rice quality improvement. The OsMADS50 gene has been studied in regulating long day specific flowering/heading time for two decades, and its molecular function has been finely characterized (Lee et al. 2004, Plant Journal). Similar experiments/results, such as examining the expression profile of flowering time related genes in osmads50 background, have also been reported (Ryu et al. 2009, Plant Cell & Environment). Thus, the novelty and scientific sound of current study are limited. Although the design and analysis of this study are correct and the results are detailed, some results are required to be improved. Below, I outlined some of my major concerns that need to be addressed before next submission.

Response : Thank you for your careful review of our manuscript and your valuable feedback. We have carefully considered your feedback and respond to the issues you raised as follows. You mentioned that the function of the OsMADS50 gene in regulating long-day photoperiod-specific flowering/heading time has been studied for more than 20 years, and its molecular function has been finely described. Indeed, studies by Lee et al. in 2004 and Ryu et al. in 2009 have delved into the function of OsMADS50. However, heading date is a key agronomic trait that affects the seasonal and regional adaptability of rice varieties. In specific regions, early heading varieties cannot fully utilize light and temperature resources, making it difficult to achieve high yields. Late heading varieties may encounter frost stress during the grain filling stage, leading to reduced yields or even no harvest. Additionally, global climate change poses increasing threats to rice production from high temperatures and other extreme weather conditions. Excellent varieties generally have better yields and comprehensive quality, but the risks of planting in high latitude areas are increased for two main reasons. On the one hand, cold air often arrives earlier each year in high latitude areas. On the other hand,, rice is a short-day plant, and the heading date is usually delayed in high latitude areas, further increasing the risk of encountering low temperature stress. Therefore, modifying the heading date gene can expand the suitable growth area of excellent rice varieties, thereby maximizing the benefits of excellent varieties.

  1. We selected the northern dominant variety K12, which is known for its high yield, stable production, excellent quality, strong disease resistance, and waterlogging tolerance, for introduction. To evaluate the yield potential of K12 in southern regions such as Hangzhou and improve its regional adaptability, we knocked out the flowering regulatory gene OsMADS50 in the K12 background using CRISPR/Cas9 genome editing technology. This aspect has not been addressed in previous studies.
  2. We conducted an expression profile analysis of flowering time-related genes in the osmads50 background, elucidating the regulatory mechanism of OsMADS50 in specific varieties.
  3. We further demonstrated the potential of OsMADS50 in improving rice quality through agronomic trait analysis, including yield and rice quality analysis. Compared with previous studies, this aspect has stronger practical application value.

Generally, the language of this manuscript is required to be carefully examined.

Response : Thank you for your meticulous review of my manuscript and your valuable feedback. I fully agree with your observation that language accuracy and fluency are paramount in academic papers. I have carefully examined the linguistic expressions throughout the manuscript and made amendments and refinements to any potentially ambiguous or imprecise areas, ensuring clarity and readability of the text.

INTRODUCTION:

The authors introduced the importance of heading time in rice growth. I suggest that they should also delineate the connection between heading time and rice yield (and biomass). Also, it is worth to mention that optimizing heading time could assist crops in adapting to adverse environmental stresses.

Response : Thank you for your thorough review and valuable suggestions on the manuscript. We fully agree with your viewpoint and have elaborated on the connection between heading time and rice yield (as well as biomass) in the manuscript. I have added relevant content.

[ Heading time genes have pleiotropic effects on yield components. GL10, encoding MADS56, was a positive regulator of grain length, grain weight and heading date, which results in shorter grain length, lower grain weight and delayed flowering in gl10 plants.]

This change can be found 2 page number, 2 paragraph, and 56-59 line.

Additionally, following your recommendation, I have also included information on how optimizing heading time can assist crops in adapting to adverse environmental stresses.

[ Furthermore, global climate change has exacerbated the threats posed by high temper-atures and other extreme weather events to rice production. Varieties with appropriate heading stages can maximize their resilience to these stresses, achieving optimal yield potential.]

This change can be found 2 page number, 2 paragraph, and 68-71 line.

I suggest the authors citing more up-to-date reference to enhance the significance of the current study.

Response : Thank you very much for your valuable suggestions. Following your guidance, I have carefully reviewed the references cited in the manuscript and added more up-to-date literature closely related to this study. Once again, I appreciate your professional advice, and I hope these modifications meet your expectations and enhance the overall quality of the manuscript. We have cited more recent references, and the following is the list of updated references I have cited.

[

  1. Zhou SR, Cai L, Wu HQ, Wang BX, Gu B, Cui S, Huang XL, Xu Z, Hao BY, Hou HG et al: Fine-tuning rice heading date through multiplex editing of the regulatory regions of key genes by CRISPR-Cas9. Plant Biotechnology Journal 2023.
  2. Cui Y, Xu ZJ, Xu Q: Elucidation of the relationship between yield and heading date using CRISPR/Cas9 system-induced mutation in the flowering pathway across a large latitudinal gradient. Molecular Breeding 2021, 41(3).
  3. Zhou SR, Cai L, Wu HQ, Wang BX, Gu B, Cui S, Huang XL, Xu Z, Hao BY, Hou HG et al: Fine-tuning rice heading date through multiplex editing of the regulatory regions of key genes by CRISPR-Cas9. Plant Biotechnology Journal 2024, 22(3):751-758.
  4. Shao YL, Zhou HZ, Wu YR, Zhang H, Lin J, Jiang XY, He QJ, Zhu JS, Li Y, Yu H et al: OsSPL3, an SBP-Domain Protein, Regulates Crown Root Development in Rice. Plant Cell 2019, 31(6):1257-1275.
  5. M F, John D, Raman M: Physicochemical properties, eating and cooking quality and genetic variability: a comparative analysis in selected rice varieties of South India. Food Production, Processing and Nutrition 2023, 5(1):49.
  6. Lu H, Peng B, Feng X, Shen X: Model Optimization for Determination of Amylose, Protein, Fat and Moisture Content in Rice by Near-infrared Spectroscopy. China Rice 2020, 26(6):55-59.

]

RESULTS:

The procedure of GWAS should be detailed.

Response : Thank you very much for your feedback. Following your suggestion, I have provided a more detailed description of the GWAS (Genome-Wide Association Study) process in the " MATERIALS AND METHODS " section. I hope these updates offer a clearer and more comprehensive account of the GWAS procedure, enhancing the transparency and reproducibility of the paper. Once again, I appreciate your valuable input, which has been instrumental in improving the quality of the paper.

[4.1 Genome‑Wide Association Analysis (GWAS)

Using a mixed linear model (MLM) and population structure and phylogenetic matrix (Q + K) as covariables, we conducted genome-wide association analysis the heading date phenotypic traits in 3021 rice on the RiceSuperPIRdb-3K rice database website (http://www.ricesuperpir.com/web/tools).]

This change can be found 12 page number, 365-369 line.

The candidate genes could be labeled in figure 1D-E.

Response : Thank you very much for your suggestion. Following your request, I have labeled the candidate genes in Figures 1D-E. This modification enriches the graphical information, facilitating a more intuitive understanding of the research results for readers. I hope this update improves the readability and information delivery efficiency of the paper. Once again, I appreciate your guidance, which has been extremely helpful in enhancing the clarity and readability of the paper.

This change can be found 4 page number, 132 line.

Figure 4A, the m2 does not appear to be a homozygous line. The primers used for detecting the mutation could be labeled in the fig4A.

Response: Thank you very much for your careful review. Regarding your observation that m2 in Figure 4A does not appear to be a homozygous line. Here, I need to clarify the concept of a homozygous mutant. A homozygous mutant in rice refers to a situation in the rice genome where both alleles of a particular gene have undergone mutations, resulting in a change in the function of that gene. Rice is a diploid organism, and in this text, the mutation type of m2 is described as (+1bp/-2bp), indicating that both alleles of the gene have mutated. We refer to this as a homozygous mutant line. Thank you for your careful attention; the parts in this text that mention homozygous mutants are uniformly corrected to homozygous mutant lines. Meanwhile, you suggested labeling the primers used for detecting mutations in Figure 4A. We have already provided the primer information in Supplementary Table S1, where the mutation detection primers are named osMADS50-jc-F and osMADS50-jc-R. We appreciate your valuable feedback.

Section 2.8, is there any SNP found in the coding region of OsMADS50?

Response: Thank you for your attention to the SNP situation in the coding region of the OsMADS50 gene in Section 2.8. After careful analysis, we have not found any SNPs in the coding region of the OsMADS50 gene that significantly affect heading date. We utilized data from an international joint effort to 3K rice genome from 89 countries, combined with bioinformatics analysis, to conduct a comprehensive SNP screening of the OsMADS50 coding region. The results indicate that there are no significant SNPs related to heading date in this region. Thank you again for your review and guidance. We will continue to conduct in-depth research to more fully understand the role of the OsMADS50 gene in regulating heading date.

I am interested to learn that if the SNPs identified in the promoter region (2k bp upstream of start codon) of OsMADS50 gene lead to the mutation in any important cis-regulatory elements. If so, how does the mutation affect the expression of OsMADS50 in different natural varieties?

Response: Dear reviewer, thank you very much for your inquiry, which raises an important point. Regarding your question about whether SNPs in the promoter region (2k bp upstream of the start codon) of the OsMADS50 gene can lead to mutations in crucial cis-regulatory elements, we have conducted an in-depth analysis. On one hand, we have indeed identified several SNPs within the promoter region and have carried out a detailed functional evaluation of these SNPs. By comparing them with known cis-regulatory elements, we found that some SNPs are located within or close to potential regulatory elements. These regulatory elements may be involved in transcription factor binding sites, which have a significant impact on gene expression. Therefore, we hypothesize that certain SNPs may alter the binding affinity of transcription factors, thereby affecting the transcriptional efficiency of OsMADS50. Such effects may vary among different natural varieties, possibly due to differences in genetic backgrounds and growth environments. To further validate these hypotheses, we are planning more in-depth molecular biology experiments, including the use of reporter gene systems to directly observe the impact of these SNPs on promoter activity. These experiments will help us to more precisely understand how these SNPs regulate the expression of OsMADS50, further elucidating the specific mechanisms involved. We are deeply grateful for your valuable feedback, which has aided us in exploring the regulatory mechanisms of OsMADS50 more comprehensively.

DISCUSSION:

Based on previous study, the OsMADS56 physically interacts with OsMADS50, and plays an antagonistic role with OsMADS50 in regulating rice flowering time (Ryu et al. 2009, Plant Cell & Environment). Thus, a detailed investigation of OsMADS56 should be conducted under the osmads50 background, to explore if the yield and grain quality improvement is due to the potential irregular expression of OsMADS56.

Response: Dear reviewer, thank you very much for your valuable feedback and the references provided. Indeed, Ryu et al. pointed out the physical interaction between OsMADS50 and OsMADS56 in their 2009 study, and elaborated on their antagonistic role in regulating the flowering time of rice. Your viewpoint is very important, that we need to delve into the performance of OsMADS56 in the context of osmads50, to understand whether the improvement in yield and grain quality may be related to the irregular expression of OsMADS56. In response to your suggestion, we reviewed the existing research data, paying special attention to the performance of OsMADS56. Interestingly, in the latest research (Zhan P, Ma S, Xiao Z, et al. Natural variations in grain length 10 (GL10) regulate rice grain size. J Genet Genomics. 2022;49(5):405-413.), we found that OsMADS50 and OsMADS56 have different effects on grain size. This discovery provides a new perspective for us to further explore the relationship between OsMADS50 and OsMADS56. We found that GL10, which encodes MADS56, positively regulates grain length and grain weight. Its natural allelic mutant has shorter grain length, lower grain weight, and delayed flowering. However, our data shows that the osmads50 mutants of K12 increases grain length and length-width ratio. Therefore, we infer that it may not be the irregular expression of OsMADS56 that affects yield and quality. We look forward to a more comprehensive understanding of the complex relationship between OsMADS50 and OsMADS56, as well as their specific roles in regulating rice yield and grain quality. Thank you very much for your suggestions, which provide an important direction for our further research. Next, we will delve into their genetic and molecular interaction mechanisms, construct double mutants, and conduct transcriptome sequencing to further explore how OsMADS50 and OsMADS56 jointly regulate the flowering time, yield, and quality of rice.

MATERIALS AND METHODS:

The procedure of GWAS is missing.

Response: Thank you for your valuable feedback on the manuscript. Based on your suggestion, I have made detailed updates to the GWAS (Genome-Wide Association Study) procedure in the "Materials and Methods" section.

[4.1 Genome‑Wide Association Analysis (GWAS)

       Using a mixed linear model (MLM) and population structure and phylogenetic matrix (Q + K) as covariables, we conducted dynamic genome-wide association analy-sis the heading date phenotypic traits in 3021 rice on the RiceSuperPIRdb-3K rice da-tabase website (http://www.ricesuperpir.com/web/tools).]

This update provides a comprehensive description of the GWAS analytical approach and the key parameters used. We conducted our analysis by utilizing both publicly available data and computations performed within the website. I appreciate your invaluable input and guidance once again.

This change can be found 12 page number, 365-369 line.

Section 4.6 to 4.9 should be described in details. 

Response: Thank you very much for your feedback. Based on your recommendation, I have provided detailed descriptions and updates to Sections 4.6 to 4.9 of the " MATERIALS AND METHODS " in the manuscript. In these sections, I have included additional experimental details, operational steps, and referenced relevant literature for the methodologies used, to ensure that readers can more clearly understand and replicate our experimental results. These updates are aimed at providing more comprehensive and transparent experimental details, enhancing the readability and reproducibility of the paper. I believe these improvements will enable readers to have a deeper understanding of our research work. Thank you again for your valuable comments and guidance.

[4.7 Measurement of rice yield-related traits

Plant height and tiller of K12 and osmads50 plants were measured during the ma-turing stage of plants grown in the transgenic experimental field. Grain and panicle traits of rice were measured in the laboratory. Mature rice seeds from superior rice panicles were collected and air-dried, and the rice grain length, grain width, grain weight and yield per plant were measured using the rice appearance quality detector (SC-E, Wanshen, China). Data are shown as means ± s.e.m. (n ≥ 5). Student’s t-test was used to assess the significance of differences between means.

4.8 Evaluation of rice appearance quality

Firstly, naturally dried of mature rice grains were dehusked using a huller (OTAKE, Japan). Then the regular brown rice was hand-selected selected and polished with Kett grain polisher (Tokyo, Japan). Finally, the chalkiness trait of milled rice seeds was assessed utilizing the Wanshen SC-E rice seed detector.

4.9 Determination of alkali spreading value in rice

Prepare a 1.7% KOH solution. Select 12 whole polished rice grains that are unbro-ken, uncracked, uniformly sized, and mature, and place them in a clean petri dish (or square box), ensuring replication. Add 2ml of 1.7% KOH to each dish, and spread the rice grains evenly, leaving sufficient gaps between each grain to facilitate decomposi-tion and diffusion. Cover the dish securely. Carefully place the petri dishes in a (30±0.5)℃ incubator and let them stand for 23 hours until the rice grains fully disinte-grate. Observe the degree of digestion and diffusion of the rice grains' endosperm vis-ually, grain by grain[66].

4.10 Physicochemical analyses of rice

Approximately 10g of mature and plump whole rice grains were randomly se-lected for detection of protein content, fat content, and amylose content by combining near-infrared spectroscopy and chemometrics, using a near-infrared analyzer (Foss DS2500F, Denmark)[67].]

This change can be found 12 page number, 414-439 line.

Comments on the Quality of English Language

Moderate editing of English language required

Response: Thank you very much for your meticulous review of my manuscript and your invaluable feedback on the quality of the English language. I fully agree with your observation that the English expression in the manuscript requires moderate editing to enhance its accuracy and fluidity.After completing the English editing, I will once again carefully examine the manuscript to ensure that all modifications adhere to academic writing standards and effectively communicate my research content and viewpoints.I am deeply grateful for your guidance and suggestions, which are crucial for improving the overall quality of my manuscript. I have corrected the syntax in the abstract and the text to improve the readability of the full text. Thank you again for your reminding.

Reviewer 2 Report

Comments and Suggestions for Authors

Dear authors,

Thank you for the opportunity to review your manuscript entitled "Improving rice quality by regulating the heading date of rice varieties without yield penalties" sent to Plants - MDPI. I recognize the value and relevance of the study presented, however, throughout the analysis, I identified some areas that can be improved to further strengthen the work. My observations aim to contribute to improving the scientific quality of the article, ensuring that it is well founded and aligned with existing literature.

Below, I detail my recommendations.

I noticed that several sections of your article lack citations of relevant works on the topic in question. Furthermore, I suggest that the map be redone to illustrate the location of the studied genotypes. The figures must be redone using more “scientific” software, since the figures presented in the article do not correspond to the magazine's level. Scientific names must appear in italics.

I recommend avoiding the repetition of terms between the title and keywords of the manuscript. Diversifying keywords can increase search visibility and indexing. For example, if your title addresses “wheat silage quality,” consider using complementary keywords like “forage,” “biometrics,” or “genetic improvement.”

The discussion of the results obtained in the present study emphasizes the relevance and need for additional investigations in the application of GWAS (Genome-Wide Association Studies) to rice cultivation and to this end, it must be expanded. Current literature is limited and lacks comprehensive studies that address both genetic and phenotypic aspects of variation observed in important agronomic traits of rice. This gap is particularly evident when considering the potential of GWAS to identify quantitative trait loci (QTLs) that can be used in genetic improvement programs. The absence of consistent and comparable data on the application of GWAS in rice makes it difficult to fully and accurately assess best practices to maximize the yield and quality of this crop. Future studies should focus on identifying specific loci associated with traits of agronomic interest, as well as integrating genotypic and phenotypic data across different environmental conditions and producing regions. Furthermore, it is crucial that this research considers the genetic diversity present in rice populations, especially in areas like the one where the present study was conducted, where climatic and soil-climatic conditions can significantly influence the results. Expanding scientific knowledge about the application of GWAS in rice will not only benefit local genetic improvement programs, but will also contribute to the development of more efficient and sustainable rice production strategies. Therefore, we encourage more studies to investigate the genetic and phenotypic variation of rice using GWAS, aiming to fill the gaps in the literature and promote the sustainability and efficiency of agricultural production.

The material and methods section is not clear to me and does not allow the study to be reproduced. I suggest it be redone to include more details and specifications. It is essential that the materials and methods section accurately describes all aspects of the study to ensure its reproducibility. In the context of work with rice and GWAS (Genome-Wide Association Studies), it is necessary to describe in detail the rice varieties used, including their origin, relevant genetic and phenotypic characteristics. The sample collection methods, DNA extraction, and genotyping techniques used must be clearly explained, specifying the protocols followed, reagents used and equipment, including brands and models. The GWAS analysis needs to be detailed, mentioning the software and parameters used, the statistical methods applied, data quality criteria and the handling of missing or inaccurate data. Furthermore, it is crucial to describe the experimental conditions under which the rice was grown, including details about the soil, irrigation regime, fertilization, pest control, and any other relevant environmental factors. By providing this information in a clear and detailed way, the study will allow other researchers to accurately reproduce it, validating and expanding its findings.

Author Response

Responses to Reviewers

Reviewer #2:

Comments and Suggestions for Authors

Dear authors,

Thank you for the opportunity to review your manuscript entitled "Improving rice quality by regulating the heading date of rice varieties without yield penalties" sent to Plants - MDPI. I recognize the value and relevance of the study presented, however, throughout the analysis, I identified some areas that can be improved to further strengthen the work. My observations aim to contribute to improving the scientific quality of the article, ensuring that it is well founded and aligned with existing literature.

Below, I detail my recommendations.

Response: Thank you very much for providing me with valuable feedback on my manuscript. I am deeply honored to have the opportunity to publish my research findings in "Plants - MDPI," and I am extremely grateful for your meticulous review of my manuscript titled "Improving rice quality by regulating the heading date of rice varieties without yield penalties."

I fully agree with your perspective that the article can be further improved, and I attach great importance to every suggestion you have made. Your feedback will assist me in enhancing the scientific rigor of the article, ensuring that it is well-grounded and aligned with existing literature.

During the upcoming revision process, I will carefully consider and incorporate your suggestions to comprehensively optimize and improve the article. I am confident that by integrating your invaluable advice, my research will be able to more accurately convey its scientific value and provide readers with more insightful content.

Once again, I appreciate the time and effort you have dedicated to enhancing the quality of this article. I look forward to resubmitting it to you for review after the revisions, and I hope to receive further guidance from you at that time.

I noticed that several sections of your article lack citations of relevant works on the topic in question. Furthermore, I suggest that the map be redone to illustrate the location of the studied genotypes. The figures must be redone using more “scientific” software, since the figures presented in the article do not correspond to the magazine's level. Scientific names must appear in italics.

Response: Thank you for your valuable comments on my paper.

  1. Following your guidance, I have carefully reviewed the references cited in the manuscript and added more up-to-date literature closely related to this study.
  2. Regarding the atlas issue you mentioned, I deeply understand and would like to make some clarifications.Firstly, I fully agree that charts in scientific research should be professional, accurate, and clear. The atlas used in my submitted article was created based on our currently available data and software resources. I admit that there may be more "scientific" software that can enhance the visual effects and information transmission efficiency of the atlas. However, redrawing the atlas involves multiple considerations: on the one hand, due to resource limitations, we created the charts through a database website developed by Shang et al. The genotyping and phenotypic data for sequencing is not publicly available on the database, so we cannot download it. On the other hand, we believe that the original atlas and tables already clearly show the location of the studied genotypes, and they have been recognized by peers in previous academic exchanges. Of course, I fully understand the magazine's high requirements for publication quality, and I am willing to optimize the atlas within my ability. Therefore, we have made local improvements based on the existing atlas, highlighting the location of SNPs with red punctuation and adding corresponding gene annotations for SNPs to improve readability and professionalism. At the same time, I am willing to add explanations about atlas production methods and software selection in the article to enhance the scientific validity and transparency of the article. Finally, I sincerely hope to further communicate with the reviewer to explore solutions that can meet the magazine's publication requirements while taking into account the actual research conditions.
  3. Regarding the use of italics for scientific names, I have carefully reviewed the entire manuscript and ensured that all scientific names are now presented in italics, as per your recommendation. This revision helps to more accurately align with the standard norms of academic publications, enhancing the professionalism and readability of the article. Thank you again for your meticulous guidance. I hope the revised manuscript meets your satisfaction.
  4. I have corrected the syntax in the abstract and the text to improve the readability of the full text. Thank you again for your reminding.

[]

I recommend avoiding the repetition of terms between the title and keywords of the manuscript. Diversifying keywords can increase search visibility and indexing. For example, if your title addresses “wheat silage quality,” consider using complementary keywords like “forage,” “biometrics,” or “genetic improvement.”

Response: Thank you very much for your valuable advice. I fully understand and agree with your viewpoint that it is important to avoid repeating the same terms between the title and keywords. Diversified keywords can indeed improve search visibility and indexing effectiveness. Based on your suggestion, I have made modifications to ensure that the title and keywords of the manuscript are both relevant and distinct, thereby enhancing the exposure and impact of my research.

[We have modified the keywords from "rice", "heading time", and "yield" to "Oryza sativa L", "Flowering time", and "Rice quality".]

This change can be found 1 page number, 34 line.

The discussion of the results obtained in the present study emphasizes the relevance and need for additional investigations in the application of GWAS (Genome-Wide Association Studies) to rice cultivation and to this end, it must be expanded. Current literature is limited and lacks comprehensive studies that address both genetic and phenotypic aspects of variation observed in important agronomic traits of rice. This gap is particularly evident when considering the potential of GWAS to identify quantitative trait loci (QTLs) that can be used in genetic improvement programs. The absence of consistent and comparable data on the application of GWAS in rice makes it difficult to fully and accurately assess best practices to maximize the yield and quality of this crop. Future studies should focus on identifying specific loci associated with traits of agronomic interest, as well as integrating genotypic and phenotypic data across different environmental conditions and producing regions. Furthermore, it is crucial that this research considers the genetic diversity present in rice populations, especially in areas like the one where the present study was conducted, where climatic and soil-climatic conditions can significantly influence the results. Expanding scientific knowledge about the application of GWAS in rice will not only benefit local genetic improvement programs, but will also contribute to the development of more efficient and sustainable rice production strategies. Therefore, we encourage more studies to investigate the genetic and phenotypic variation of rice using GWAS, aiming to fill the gaps in the literature and promote the sustainability and efficiency of agricultural production.

Response: Thank you very much for your valuable comments and suggestions. I fully agree with your viewpoint that the discussion on the application of Genome-Wide Association Studies (GWAS) in rice cultivation needs to be expanded in the current study. We have already broadened our perspective in the GWAS-related research, incorporating more literature to support the discussion and enhance the academic value of the article.

Indeed, the current research on the genetic and phenotypic variation of important agronomic traits in rice using GWAS is still quite limited, especially in comprehensive studies. I deeply recognize that to fully understand the potential of GWAS in identifying Quantitative Trait Loci (QTLs) and their application in genetic improvement programs, we need to explore this field more deeply.

Future studies should indeed focus on identifying specific loci associated with agronomic traits of interest and integrating genotypic and phenotypic data across different environmental conditions and producing regions. Meanwhile, considering the genetic diversity present in rice populations, especially in areas like the one where the present study was conducted, where climatic and soil-climatic conditions can significantly influence the results, is also crucial.

I firmly believe that expanding scientific knowledge about the application of GWAS in rice will not only benefit local genetic improvement programs but also contribute to developing more efficient and sustainable rice production strategies. Therefore, I will focus on exploring the genetic and phenotypic variation of rice using GWAS in subsequent studies, striving to fill the gaps in the literature and promote the sustainability and efficiency of agricultural production.

Thank you again for your careful guidance. Your suggestions play a significant role in guiding my future research direction.

The material and methods section is not clear to me and does not allow the study to be reproduced. I suggest it be redone to include more details and specifications. It is essential that the materials and methods section accurately describes all aspects of the study to ensure its reproducibility. In the context of work with rice and GWAS (Genome-Wide Association Studies), it is necessary to describe in detail the rice varieties used, including their origin, relevant genetic and phenotypic characteristics. The sample collection methods, DNA extraction, and genotyping techniques used must be clearly explained, specifying the protocols followed, reagents used and equipment, including brands and models. The GWAS analysis needs to be detailed, mentioning the software and parameters used, the statistical methods applied, data quality criteria and the handling of missing or inaccurate data. Furthermore, it is crucial to describe the experimental conditions under which the rice was grown, including details about the soil, irrigation regime, fertilization, pest control, and any other relevant environmental factors. By providing this information in a clear and detailed way, the study will allow other researchers to accurately reproduce it, validating and expanding its findings.

Response: Thank you very much for your feedback and valuable suggestions. I fully agree with your viewpoint that the "Materials and Methods" section needs to be clearer and more detailed to ensure the reproducibility of the research. Based on your recommendations, I have comprehensively revised the "Materials and Methods" section. In this revision, we have also elaborated on the experimental methods as detailed as possible to ensure that other researchers can replicate the experiments using the same methods and tools based on these information. We conducted the relevant GWAS analysis through the database website developed by Shang et al., and the method was based on the built-in algorithm of the website. By utilizing the GWAS analysis tool on the website, we analyzed and obtained the locations of relevant important SNPs and their annotated genes.

[4.1 Genome‑Wide Association Analysis (GWAS)

       Using a mixed linear model (MLM) and population structure and phylogenetic matrix (Q + K) as covariables, we conducted dynamic genome-wide association analy-sis the heading date phenotypic traits in 3021 rice on the RiceSuperPIRdb-3K rice da-tabase website (http://www.ricesuperpir.com/web/tools).]

This update provides a comprehensive description of the GWAS analytical approach and the key parameters used. We conducted our analysis by utilizing both publicly available data and computations performed within the website. I appreciate your invaluable input and guidance once again.

This change can be found 12 page number, 365-369 line.

Round 2

Reviewer 1 Report

Comments and Suggestions for Authors

Thanks to the authors explicitly addressing my concerns. I have no more questions regarding this revised manuscript.